# Universal embodied intelligence: learning from crowd, recognizing the world, and reinforced with experience

## Abstract

The interactive artificial intelligence in the motion control field is an interesting topic, especially when universal knowledge adaptive to multiple task and universal environments is wanted. Although there are increasing efforts on Reinforcement learning (RL) studies with the assistance of transformers, it might subject to the limitation of the offline training pipeline, in which the exploration and generalization ability is prohibited. Motivated by the cognitive and behavioral psychology, such agent should have the ability to learn from others, recognize the world, and practice itself based on its own experience. In this study, we propose the framework of Online Decision MetaMorphFormer (ODM) which attempts to achieve the above learning modes, with a unified model architecture to both highlight its own body perception and produce action and observation predictions. ODM can be applied on any arbitrary agent with a multi-joint body, located in different environments, trained with different type of tasks. Large-scale pretrained dataset are used to warmup ODM while the targeted environment continues to reinforce the universal policy. Substantial online experiments as well as few-shot and zero-shot tests in unseen environments and never-experienced tasks verify ODM's performance, and generalization ability. Our study shed some lights on research of general artificial intelligence on the embodied and cognitive field studies. Code, result and video examples can be found on the website https://baimaxishi.github.io/.

## 1 Introduction

Research of embodied intelligence focus on the learning of control policy given the agent with some morphology (joints, limbs, motion capabilities), while it has always been a topic whether the control policy should be more general or specific. As the improvement of large-scale data technology and cloud computing ability, the idea of artificial general intelligence (AGI) has received substantial interest (Reed et al., 2022). Accordingly, a natural motivation is to develop a universal control policy for different morphological agents and easy adaptive to different scenes. It is argued that such a smart agent could be able to identify its 'active self' by recognizing the egocentric, proprioceptive perception, react with exteroceptive observations and have the perception of world forward model (Hoffmann & Pfeifer, 2012). However, there is seldom such machine learning framework by so far although some previous studies have similar attempts in one or several aspects.

Reinforcement Learning(RL) learns the policy interactively based on the environment feedback therefore could be viewed as a general solution for our embodied control problem. Conventional RL could solve the single-task problem in an online paradigm, but is relatively difficult to implement and slow in practice, and lack of generalization and adaptation ability. Offline RL facilitates the implementation but in cost of performance degradation. Inspired by recent progress of large model on language and vision fields, transformer-based RL (Reed et al., 2022; Chen et al., 2021; Lee et al., 2022; Janner et al., 2021; Zheng et al., 2022; Xu et al., 2022) has been proposed by transforming RL trajectories as a large time sequence model and train it in the auto-regressive manner. Such methods provide an effective approach to train a generalist agent for different tasks and environments, but usually have worse performance than classic RL, and fail to capture the morphology information. In contrast, MetaMorph (Gupta et al., 2022) chooses to encode on agent's body mor-

phology and performs online learning, therefore has good performance but lack of time-dependency consideration.

To have a better solution of embodied intelligence, we are motivated from behavioral psychology in which agent improve its skill by actual practice, learning from others (teachers, peers or even someone with worse skills), or makes decision based on the perception of 'the world model' (Ha & Schmidhuber, 2018; Wu et al., 2022). It is reasonable to believe that an embodied intelligence agent should have the above three learning paradigm simultaneously. We propose such a methodology by designing a morphology-time transformer-based RL architecture which is compatible with both offline and online learning. Offline training is conducted on multi-task datasets which considers both learning from other agents and speculate the future system states. The online training allows the agent to improve its policy in an on-policy way given a single task.

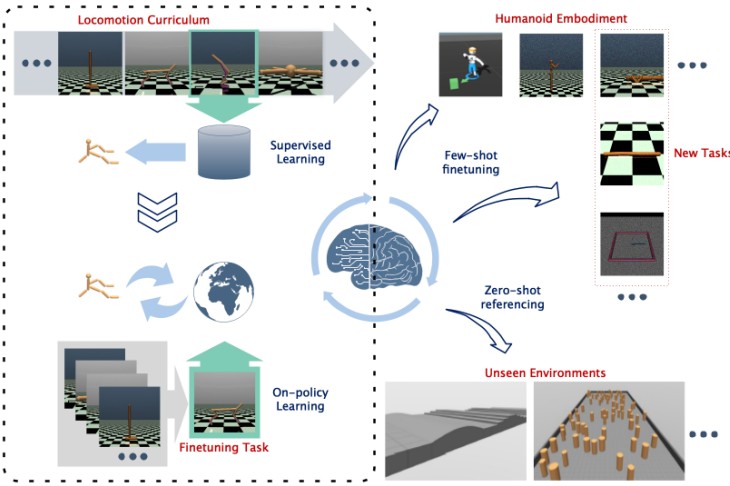

Figure 1: Application pipeline of ODM.

In this work, we propose a framework called **O**nline **D**ecision **M**etamorphformer (ODM), which aims to study the general knowledge of embodied control across different body shapes, environments and tasks, as indicated in Figure 1. The model architecture contains the universal backbone and the task-specific modules. The task-specific modules capture the potential difference in agent body shapes, and the morphological difference is enhanced by a prompt based on characteristic of body shapes. We first pretrain this model with a curriculum learning, by learning demonstrations from the easiest to the hardest task, from the expert to low-level players. The environment model prediction is added as an auxiliary loss. The same architecture can then be finetuned online given a specific task. During the test, we are able to test ODM with all training environments, transfer the policy to different body shapes, adaptive to unseen environments and accommodate with new types of tasks (e.g. from locomotion to reaching, target capturing or escaping from obstacles.).

Main contributions of this paper include:

- We design a unified model architecture to encode time and morphology dependency simultaneously which bridges sequential decision making with embodiment intelligence.
- We propose a training paradigm which mimic the process of natural intelligence emerging, including learning from others, boost with practices, and recognize the world.
- We train and test our framework with agent in eight different body shapes, different environment terrain and different task types. These comprehensive analysis verifies the general knowledge of motion control learned by us.

## 2 RELATED WORKS

**Classic RL**: Among conventional RL methods, on-policy RL such as Proximal Policy Optimization (PPO (Schulman et al., 2017) is able to learn the policy therefore has good adaptive ability to environment, but is slow to convergence and might have large trajectory variations. Off-policy RL

such as DQN (Mnih et al., 2015) improves the sampling efficiency but still require the data buffer updated dynamically. In contrast, offline RL (Fujimoto et al., 2019; Kumar et al., 2020; Kostrikov et al., 2021) can solve the problem similar with supervised learning, but might have degraded performance because of the distribution shift between offline dataset and online environment. In our work, we aim to reach the state-of-the-art performance for different embodied control tasks, therefore a model architecture compatible with on-policy Rl is proposed.

**Transformer-based RL**: Among these efforts, Decision Transformer (DT) (Chen et al., 2021) and Multi-game Decision Transformer (Lee et al., 2022) embodied the continuous state and action directly and use a GPT-like casual transformer to solve the policy offline. Their action decision is conditioned on Return-to-Go (RTG), either arbitrarily set or estimated by model, since RTG is unknown during inference. Instead, Trajectory Transformer (TT) (Janner et al., 2021) discards RTG in the sequential modeling to avoid that approximation.PromptDT (Xu et al., 2022) adds the task difference consideration into the model by using demonstrated trajectory as prompts. ODT (Zheng et al., 2022) first attempts to solve transformer-based RL in an online manner but mainly focus on supervised on actions instead of maximizing rewards. In our work, we propose a similar model architecture but is able to conduct both offline learning and on-policy, actor-critic learning. The online learning employs PPO as the detailed policy update tools with the objective as reward maximization.

**Morphology-based RL**: There are some other studies which focus on the agent's morphology information, including GNN-based RL which modeling agent joints as a kinematics tree graph (Huang et al., 2020), Amorpheus (Kurin et al., 2021) which encodes a policy modular for each body joint, and MetaMorph (Gupta et al., 2022) which intuitively use transformer to encode the body morphology as a sequence of joint properties, and train it by PPO. In our work, we have the morphology-aware encoder which is similar with MetaMorph and the same PPO update rule. However, comparing with Metamorph, we encode the morphology on not only state but also historical actions, and consider the historical contextual consideration.

## 3 PRELIMINARIES AND PROBLEM SETUP

We formulate a typical sequential decision making problem, in which on each time step $t$, an embodied agent conceives a state $s_t \in \mathcal{R}^{n_s}$, performs an action $a_t \in \mathcal{R}^{n_a}$, and receives a scalar reward $r_t \in \mathcal{R}^1$. **Reinforcement Learning (RL)** can then be employed to produce the policy $\pi(a_t|s_t)$ which aims to maximize the expectation sum of discounted rewards. The actor-critic framework is a famous RL framework with the critic estimates the state value function $V(s)$, which the actor determines the policy. Classical RL methodologies such as Proximal Policy Optimization (PPO) can be employed to solve the problem effectively, with detailed derivation in Appendix A.

**Problem Setup**: Here we redefine the aforementioned conventional RL notations in a more 'embodied style', although still generalized enough for any arbitrary agent with multi-joint body. Inspired by the idea of Gupta et al. (2022), we differentiate the observation into the agent's proprioceptive observations, the agent's embodied joint-wise self-perception (e.g. angular, angular velocity of each joint), as well as the exteroceptive observation, which is the agent's global sensory (e.g. position, velocity). Given a $K$-joint agent, we denote the proprioceptive observation by $o^{pro} \in \mathbb{R}^{K \times n}$ in which each joint is embedded with $n$ dimension observations. The exteroceptive observation is $x$-dimensional which results in $o^{ext} \in \mathbb{R}^x$ and $s := [o^{pro}, o^{ext}]$.

Table 1: Comparison of conventional RL and our notations

| | State | Action |
|---|---|---|
| Conventional RL | $s \in \mathcal{R}^{n_s}$ | $a \in \mathcal{R}^{n_a}$ |
| Our approach | $o^{pro}, \mathcal{M}_s^{pro} \in \mathbb{R}^{K \times n}, o^{ext} \in \mathbb{R}^x$ $s = [o^{pro}, o^{ext}]$ | $a, \mathcal{M}_a \in \mathbb{R}^{K \times m}$ |
| Connections | $n_s = K * n - \sum \mathcal{M}_s + x$ | $n_a = K * m - \sum \mathcal{M}_a$ |

Stepping forward from Gupta et al. (2022), we also define the action in the joint-dependent way; that is, assuming each joint has $m$ degree of freedom (DoF) of movements (e.g. torque), the action is reshaped as $a \in \mathbb{R}^{K \times m}$. To allow the room of different agent body shapes, we introduce binary masks which have the same shapes of $o^{pro}$ and $a$ and zero-pad the impossible observations or actions (e.g. DoF of a humanoid's forearm should be smaller than its upper-arm due to their physical connection). Table 1 visualizes the comparison between the conventional RL notations and our embodied version notations.

**Attention-based Encoding**: Given a stacked time sequence vector $x \in \mathbb{R}^{T \times e}$ with $T$ as the time length and $e$ as the embedding dimension, an time sequence encoder can be expressed as

$$\text{Enc}_T(x) = \text{Attention}(\text{Q=}x, \text{K=}x, \text{V=}x) \in \mathbb{R}^{T \times e} \tag{1}$$

according to the self-attention mechanism (Vaswani et al., 2017) with $Q, K, V$ denoting query, key and value. Analogously, given a stacked joint sequence vector $p \in \mathbb{R}^{K \times e}$ with $K$ as the number of joints, a morphology-aware encoder instead learns the latent representation on this joint sequence

$$\text{Enc}_M(p) = \text{Attention}(\text{Q=}p, \text{K=}p, \text{V=}p) \in \mathbb{R}^{K \times e} \tag{2}$$

By pre-tokenizing either $o^{pro}$ or $a$ into $p$, within the latent space with dimension $e$, their 'pose' latent variables can be encoded by $\text{Enc}_M$. For each form of encoder, timestep or joint position info is encoded by lookup embedding then adding to encoded vector. More details can be referred to the MetaMorph paper (Gupta et al., 2022).

## 4 METHOD

### 4.1 MODEL ARCHITECTURE

Our ODM model structure contains a unified backbone and task-specific modules. Within this paper's context, task might be related with different agent (potentially different body types), environments and reward mechanism. Since the original system variables might have different dimensions, task-specific modules map them into a uniform-dimensional latent space ($e$ in Eq 2) and reverse operations. The backbone has a two-directional, morphology-time transformer structure, including morphology-aware encoders and a casual transformer. Architecture details are specified in Fig 2.

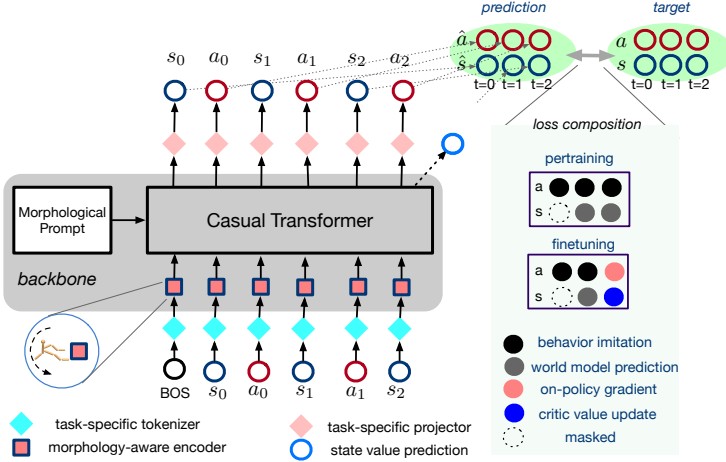

Figure 2: Model structure of ODM and its training paradigm.

**Tokenizer**: At each time $t$, observations and actions are first embedded into the latent space

$$o_t^e = \text{Embed}_o(o_t^{pro}) \in \mathbb{R}^{K,e}, \quad x_t^e = \text{Embed}_x(o_t^{ext}) \in \mathbb{R}^e, \quad a_t^e = \text{Embed}_a(a_t) \in \mathbb{R}^{K,e} \tag{3}$$

**Morphology-aware Encoder**: Corresponding pose embedding vectors are obtained by traversing the agent's kinematic tree and encoding the morphology by Eq 2:

$$o_t^p = \text{Enc}_M(o_t^e), \quad s_t^p = \text{MLP}_s([s_t^p, x_t^e]) \in \mathbb{R}^e, \quad a_t^p = \text{Enc}_M(a_t^e) \tag{4}$$

**Casual Transformer**: To capture the morphology difference, we apply the prompt technique as in (Xu et al., 2022), but embedding the morphology specifications instead of imitations

$$\text{Prompt} = \text{Embed}(K, n, m, x) \tag{5}$$

The casual transformer then translates the prompt and the input sequence into the output sequence

$$\text{output} = \text{Enc}_T(\text{Prompt}, \text{input}) \tag{6}$$

$$\text{input} := \{\text{BOS}, s_0^p, \quad a_0^p, s_1^p, \quad \cdots, \cdots, \quad a_{t-1}^p, s_t^p\} \tag{7}$$

$$\text{output} := \{\hat{s}_0^p, \hat{a}_0^p, \quad \hat{s}_1^p, \hat{a}_2^p, \quad \cdots, \cdots, \quad \hat{s}_t^p, \hat{a}_t^p\} \tag{8}$$

with a forward casual time mask. Detailed structure is inherited from GPT2, a decoder-only structure as in (Chen et al., 2021; Janner et al., 2021; Zheng et al., 2022).

**Projector**: The task-specific projectors map latent outputs back to the original spaces:

$$\hat{a}_t = \text{Proj}_a(\hat{a}_t^p), \quad \hat{s}_t = \text{Proj}_s(\hat{s}_t^p), \quad \hat{V}_t = \text{Proj}_V(\hat{s}_t^p) \tag{9}$$

$\text{Embed}_o, \text{Embed}_x, \text{Embed}_a, \text{Embed}_s, \text{Proj}_a, \text{Proj}_s, \text{Proj}_V$ are all modeled as MLPs with LayerNorm and Relu between layers. More detailed configuration can be found in Appendix.

## 4.2 TRAINING PARADIGM

ODM has a two-phase training paradigm including pretraining and finetuning, as in Algorithm 1.

**Pretraining**: To mimic the learning process of human infant, we design a curriculum-based learning mechanism in which the training dataset transverses from the easiest to the most complicated one. During each epoch, the current dataset is trained in a auto-regressive manner with two loss terms:

$$L^{\text{imitation}} = \text{MSE}(\hat{a}_t, a_t^p), \quad L^{\text{prediction}} = \text{MSE}(\hat{s}_t, s_t^p), \quad L^{\text{pretrain}} = \eta_p L^{\text{imitation}} + \eta_i L^{\text{prediction}} \tag{10}$$

where MSE denotes the mean-square-error. $L^{\text{imitation}}$ corresponds the imitation of action from demonstrations, while $L^{\text{prediction}}$ encourages the agent to predict future observations [1].

**Finetuning**: one extra predict head is activated to predict the state value $\hat{V}_t$; this head as long as the very last prediction head of $\hat{a}_t$ are employed as outputs of actor and critic:

$$\hat{V}_t \to V(s_t), \quad \hat{a}_t \to \pi(s_t) \tag{11}$$

Actor and critic are then trained by PPO with more details introduced in Section A. Keeping some extent of $L^{\text{pretrain}}$ as auxiliary loss, this finetuning becomes a self-supervised plus model-based RL

$$L^{\text{finetune}} = \eta_1 L^{\text{PPO}} + \eta_2 L^{\text{pretrain}} \tag{12}$$

---

**Algorithm 1** ODM

1: **Initialize** $\theta$
2: **Pretraining:**
3:     set num of epoch = 0
4:     **SWITCH** between 6 body shapes :
5:         activate grad of env-specific module of current env, freeze others
6:         **REPEAT** learning from pioneers of different expert levels
7:             self-regressive training on $L^{\text{pre}}$ of a mini-batch
8:         increment num of epoch
9:     save the model checkpoint
10: **Finetuning:**
11:     load the model in the target environment
12:     **REPEAT** iterations:
13:         **for** each actor **do**
14:             run current policy $\pi$ for $T$ timesteps
15:             compute advantage estimates $\hat{A}_0, \cdots, \hat{A}_T$
16:         **end for**
17:         update $\theta$ with the surrogate $L^{\text{PPO}}$ on a mini-batch
18:         stop when converges

---

[1] $L^{\text{prediction}}(t = 0)$ is masked out, since it is meaningless to predict the very first, randomly initiated state.

## 5 EXPERIMENTS

### 5.1 EXPERIMENT CONFIGURATIONS

**Bodies, Environments and Tasks**: We practice with enormous agents, environments and tasks, to validate the general knowledge studied by ODM. These scenes include:

- *Body shape*: including swimmer (3-joints, no foot), reacher (1-joint and one-side fixed), hopper (1 foot), halfcheetah (2-foot), walker2d (2-foot), ant (4-foot), and humanoid on the gym-mujoco platform [2]; walker (the agent called ragdoll has a realistic humanoid body) [3] on the unity platform (Juliani et al., 2018); and finally unimal, a mujoco-based enviroment which contain 100 different morphological agents (Gupta et al., 2021).
- *Environment*: flat terrain (FT), variable terrain (VT) or escaping from obstacles.
- *Task*: pure locomotion, standing-up (humanoid), or target reaching (reacher, walker).

**Baselines**: We compare ODM with four baselines, each representing a different learning paradigm:

- *Metamorph*: a morphological encoding-based online learning method to learn a universal control policy (Gupta et al., 2022).
- *DT*: As a state-of-the-art offline learning baseline, we implement the decision transformer with the expert action inference as in Lee et al. (2022) and deal with continuous space as in (Chen et al., 2021). We name it DT in the following sections for abbreviation.
- *PPO*: The classical on-policy RL algorithm Schulman et al. (2017). Code is cloned from stable-baseline3 in which PPO is in the actor-critic style.
- *Random*: The random control policy by sampling each action from uniform distribution from its bounds. This indicates the performance without any prior knowledge especially for the zero-shot case.

**Demonstrating pioneers**: For purpose of pretraining, we collect offline data samples of hopper, halfcheetah, walker2d and ant from D4RL (Fu et al., 2020), as sources of pioneer demonstrations. For these environments, D4RL provide datasets sampled from agents of different skill levels, which corresponds to different learning pioneers in our framework, including *expert*, *medium-expert*, *medium*, *medium-replay* and *random*. We also train some baseline expert agents and using them to sample offline dataset on walker and unimal. These dataset contains more than 25 million data samples, with statistics details shown in Appendix, Table 8. Within each curriculum, we also rotate demonstrations from the above pioneers for training, as indicated in Algorithm 1.

### 5.2 EXPERIMENT RESULTS

**Pretraining**: Model is trained with datasets of hopper, halfcheetah, walker2d, ant, walker and unimal, from the easiest to the most complex. Figure 3 shows the loss plot. One can observe that the training loss successfully converges within each curriculum course; although its absolute value occasionally jumps to different levels because of the environment (and the teacher) switching. Validation set accuracy is also improved with walker and unimal as exhibition examples in Figure 3.

**Online Experiments**: To make the online learning faster, we use 32 independent agents to sample the trajectory in parallel, with 1000 as the maximum episode steps. Experiment continues more than 1500 iterations after the performance converges. Figure 4 provides a quick snapshot of online performances. Comparing with ODM without pretraining, returns of ODM are higher at the very beginning, indicating knowledge from pretraining is helpful. As online learning continuous, the performance degrades slightly until finally grows up again, and converges faster than the other two methods, although the entire training time (pretraining plus finetuning) is longer.

During online testing, 100 independent episodes are sampled, and each is fed with a unique random seed. We examine the averaged episode return and episode length as our evaluation metrics . Table 2 shows the online testing performance. One can observe that our ODM outperforms or is at

---

[2]https://www.gymlibrary.dev/environments/mujoco/
[3]https://github.com/Unity-Technologies/ml-agents

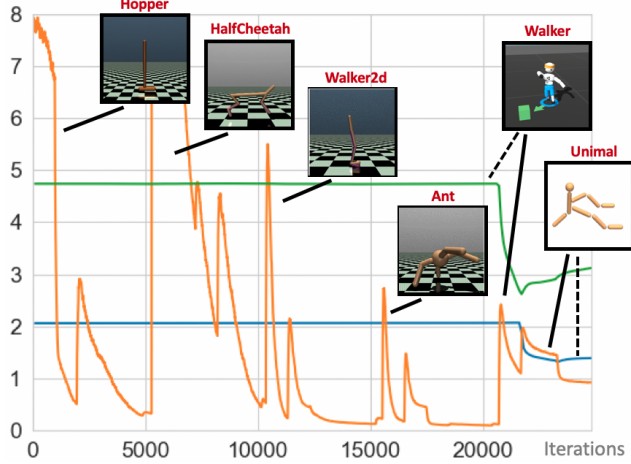

Figure 3: Time plot of pretraining performance. Orange: training loss. Green: validation MSE of walker; Blue: validation MSE of unimal.

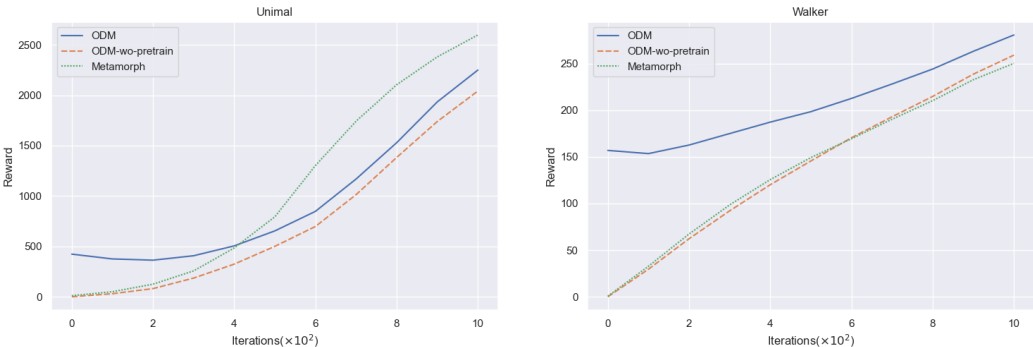

Figure 4: Comparison of averaged episode returns during online experiments. Left: unimal. Right: walker. Curves are smoothed and values are rescaled for better visualization.

least similar with Metamorph's performance. It is worth noting that DT does not work for unimal, indicating the limitation of pure offline method with changing agent body shapes.

Table 2: Averaged Episodic Performance in online locomotion environments.

| Metric | Env. | ODM | Metamorph | DT | PPO | Random |
|--------|------|-----|-----------|-----|-----|--------|
| return | walker* | **331.88**±280.96 | 303.933±279.16 | 252.74±281.18 | 265.23±275.32 | 0.55±0.83 |
|        | unimal | 3197.22±228.04 | **3251.68**±192.61 | -0.09±3.75 | 2507.32±260.71 | -3.54±4.97 |
| length | walker* | **133.29**±35.88 | 128.42±33.52 | 126.37±33.73 | 128.56±34.06 | 10.34±1.55 |
|        | unimal | 917.85±40.84 | **931.92**±33.21 | 347.36±66.47 | 884.39±50.34 | 321.98±71.49 |

∗: Performance of walker has substantial deviations since walker has forward process noise implemented.

**Few-shot Experiments**: We examine the policy transferability by providing several few-shot experiments. Pretrained ODM is loaded in several unseen tasks, which are listed in Table 3. As a few-shot test, online training only lasts for 500 steps before testing. ODM obtains the best performance except humanoid on flat terrain, indicating ODM has better adaptation ability than MetaMorph.

**Zero-shot Experiments**: Zero-shot experiments can be conducted by inferencing the model directly without any online finetuning. The unimal environment allows such experiment in which the flat terrain (FT) can be replaced by variable terrain (VT) or obstacles. Results are shown in Table 4. It can be observed that ODM reaches state-of-the-art performance for zero-shot tests, indicating that ODM has strong generalization ability by capturing general high-level knowledge from pretraining, even without any prior experience.

Table 3: Performance in few-shot experiments.

| Metric | Env. | Task | ODM | Metamorph | PPO | Random |
|---|---|---|---|---|---|---|
| return | unimal | obstacle | **1611.86**±179.38 | 1288.15±127.48 | 932.34±79.45 | -2.08±4.85 |
| | unimal | VT | **580.10**±41.23 | 499.58±35.21 | 310.02±22.93 | -4.87±8.19 |
| | swimmer | FT | **145.58**±16.97 | 143.01±11.31 | 142.36±13.31 | 0.14±2.00 |
| | reacher | target reaching | **-32.97**±5.18 | -33.28±4.58 | -34.28±4.56 | -42.96±0.15 |
| | humanoid | FT | 359.90±54.84 | 360.87±51.28 | **360.70**±50.28 | 108.13±0.83 |
| | humanoid | standup | **76388.12**±906.01 | 75750.41±897.01 | 75033.74±895.20 | 38921.67±451.25 |
| length | unimal | obstacle | **827.85**±53.25 | 771.38±50.18 | 619.82±68.35 | 322.05±65.81 |
| | unimal | VT | **780.23**±89.02 | 764.48±77.73 | 524.10±59.92 | 542.70±88.76 |
| | swimmer | FT | 1000 | 1000 | 1000 | 1000 |
| | reacher | reaching | 50* | 50* | 50* | 50* |
| | humanoid | FT | **68.10**±3.15 | 66.85±3.94 | 67.59±2.34 | 22.15±0.17 |
| | humanoid | standup | 1000 | 1000 | 1000 | 1000 |

∗: The official reacher environment has a maximum episode length limit of 50.

Table 4: Performance in zero-shot experiments.

| Metric | Env. | Task | ODM | Metamorph | DT | Random |
|---|---|---|---|---|---|---|
| return | unimal | obstacle | **1271.70**±182.34 | 1137.52±178.60 | -0.55±3.06 | -2.08±4.85 |
| | unimal | VT | **521.08**±34.48 | 480.29±23.21 | 8.22±6.70 | -4.87±8.19 |
| length | unimal | obstacle | **750.99**±86.23 | 736.90±75.16 | 228.20±55.74 | 322.05±65.81 |
| | unimal | VT | **698.80**±69.49 | 664.63±72.25 | 585.13±83.54 | 542.70±88.76 |

**Ablation Studies**: To verify the effectiveness of each model component, we conduct the ablation tests for ODM with only online finetuning phase (wo pretrain) and with only pretraining (wo finetune); within the pretraining scope, we further examine ODM without the curriculum mechanism (wo curriculum) and morphology prompt (wo prompt). The DT method could be viewed as the ablation of both $L^{\text{prediction}}$ and $L^{\text{PPO}}$, so we do not list the ablation results of these two loss terms. We conduct the ablation study on unimal (all 3 tasks) as well as walker, with results shown in Table 5. Results shown that ODM is still the best on all these tasks, which indicating both learning from others' imitation and self-experiences are necessary for intelligence agents.

Table 5: Performance in ablation studies on unimal and walker.

| Metric | Env. | Task | ODM | wo pretrain | wo finetune | wo curriculum | wo prompt |
|---|---|---|---|---|---|---|---|
| return | unimal | FT | **3197.22**±228.04 | 2331.36±131.24 | 463.12±84.55 | 434.44±73.43 | 453.41±80.13 |
| | unimal | obstacle | **1611.86**±179.38 | 592.96±89.92 | 80.65±34.91 | 78.34±32.52 | 75.23±32.21 |
| | unimal | VT | **580.10**±41.23 | 404.68±122.46 | 70.23±31.31 | 69.34±30.34 | 70.01±32.05 |
| | walker | FT | **331.88**±280.96 | 313.49±260.35 | 112.64±72.23 | 109.34±68.19 | 111.82±69.76 |
| length | unimal | FT | **917.85**±40.84 | 845.78±79.55 | 436.46±70.0 | 433.24±70.0 | 436.46±70.0 |
| | unimal | obstacle | **827.85**±53.25 | 554.89±59.02 | 232.12±37.02 | 239.12±29.13 | 230.78±35.98 |
| | unimal | VT | **780.23**±89.02 | 526.22±34.61 | 209.75±35.41 | 205.51±33.34 | 204.14±30.61 |
| | walker | FT | **133.29**±35.88 | 105.29±40.13 | 84.32±34.17 | 80.41±33.41 | 82.14±35.43 |

**Typical Visualizations**: Generalist learning not only aims to improve the mathematical metrics, but also the motion reasonability from human's viewpoint. It is difficult for traditional RL to work on this issue which only solve the mathematical optimization problem. By jointly learning other agent's imitation and bridge with the agent's self-experiences, we assume ODM could obtain more universal intelligence about body control by solving many different types of problems. Here we provide some quick visualizations about generated motions of ODM, comparing with the original versions [4].

By examining the agent motion's rationality and smoothness, we first visualize motions of trained models on the walker environment. Since the walker agent has a humanoid body (the 'ragdoll') such that reader could easily evaluate the motion reasonability based on real life experiences. Figure 5 exhibit key frames of videos on the same time points. In this experiment, we force the agent starts from exact the same state and remove the process noise. By comparing ODM (the bottom line) with PPO (the upper line), one can see the ODM behaves more like human while PPO keep swaying forwards and backwards and side to side, with unnatural movements such as lowering shoulders and twisting waist.

We compare the motion agility by visualizing on the unimal environment, in which the agent is encouraged to walk toward arbitrary direction. Figure 6 comparing ODM with Metamorph. Metamorph wastes most of time shaking foots, fluiding and gliding, therefore ODM walks longer distance than the Metamorph, within the same time interval [5].

---

[4] Full version of videos can be found on the website `https://baimaxishi.github.io/`

[5] Figure grids could help reader to recognize the comparison although the video is more obvious.

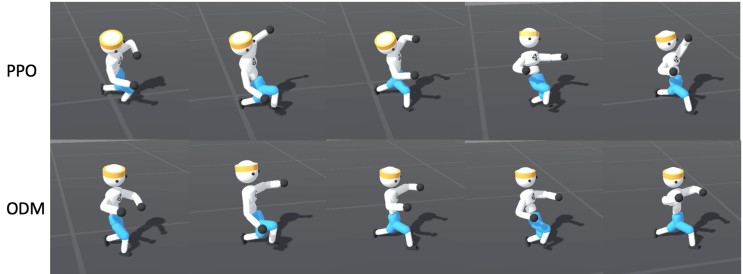

Figure 5: ODM improve motion fluency and coherence in the walker environment. Key frames are screened on Second 1, 2, 3, 4, 5, respectively. Video can be found on the website.

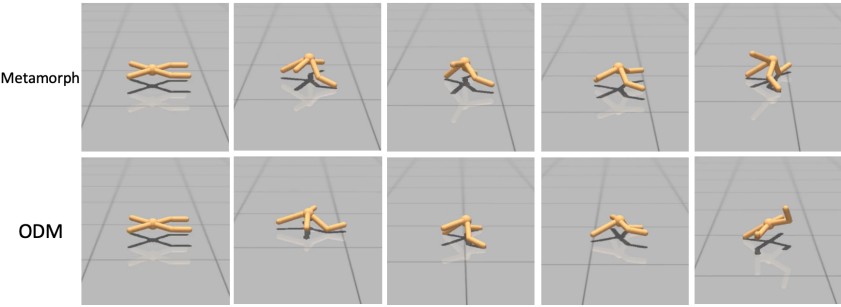

Figure 6: ODM improve motion agility of a typical unimal agent. Key frames are screened evenly from a 30-second video. Video can be found on the website.

## 6 DISCUSSION

Our work can be viewed as an early attempt of an embodied intelligence generalist accommodated for varied body shape, tasks and environment. One shortcoming of current approach is that ODM still has task-specific modules (tokenizers and projectors) for varied body shapes. By using some self-adaptive model structure (e.g. Hypernetwork) in these modules, it is possible to use one unified model to represent the generalist agent. Another potential improvement is to add the value/return prediction into the sequence modeled by the casual transformer. That is, the agent is able to estimate 'the value of its action' before the action is actually conducted, which is also known as 'metacognition'. The last interesting topic is the potential training conflict when training switch from offline to online. That might be improved by some hyperparameter tuning (out of this paper's scope), e.g., some warmup schedule of $L^{\text{PPO}}$ during finetuning; but could also be improved by different model architecture which could better accommodate knowledge learning from offline and online phases.

## 7 CONCLUSION

In this paper, motivated by the intelligence development process in the natural world, we propose a learning framework to learn a universal body control policy in arbitrary body shapes, environments and tasks. We combine ideas of learning from others, reinforcing with self-experiences, as well as the world model recognition. To achieve this, we design a two-dimensional transformer structure which first encode the morphological information of agent states and actions at each time step, then encode the time sequential decision process to formulate the policy function. A two-phase training paradigm is designed accordingly in which the agent first learns from demonstrations of pioneers on different skill levels, and from different tasks. After that, the agent interact with its own environment and further reinforce its skill by on-policy RL. Online, few-shot and zero-shot experiments show that our methodology is able to learn some general knowledge for embodied motion control. We believe this work could shed some light on the embodied intelligence study when some kind of generalist intelligent is wanted.

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
