# OpenReview forum: "Universal embodied intelligence: learning from crowd, recognizing the world, and reinforced with experience"
_ICLR.cc/2023/Conference — Submitted to ICLR 2023_

### Official Review · Reviewer_Vvnb · 2022-10-22

**Confidence:** 4
**Correctness:** 2
**Technical Novelty And Significance:** 2
**Empirical Novelty And Significance:** 1
**Recommendation:** 1

**Clarity, Quality, Novelty And Reproducibility:**

* Clarity: This paper needs a major revision. There are many grammatical errors, unclear notations, and unclear explanations. Some examples are the following.
  * Why is it called Morphology-aware "Auto-Encoder" when there is no reconstruction loss?
  * What does $+$ symbol mean in Equation 15?
  * $L^{prediction}$ seems like a reconstruction loss. Why would it encourage the agent to learn a "world model"?
  * X-axis is not defined in the plots.
  * Error bars are missing in the plots.

* Quality: Many important experiments are missing as pointed out above.

* Novelty: Although the particular architecture is new, the idea of learning a universal architecture that can handle multiple morphologies and tasks through supervised learning (e.g., GATO) and reinforcement learning (e.g., MetaMorph) has been explored.

* Reproducibility: The paper described hyperparameters in the appendix but not the code.

**Strength And Weaknesses:**

[Strength]
* Proposes a potentially interesting architecture.

[Weakness]
* The presentation of this paper needs to be improved significantly. (see the next section).
* The improvement over the baselines (especially MetaMorph) is not statistically significant (Error bars highly overlap).
* Many important experiments are missing. For example, there is experiment showing generalization to new morphologies, which is important for supporting the main claim of the paper. There is no ablation for the use of curriculum in the pre-training phase, the use of morphology prompt, and the use of each loss term (e.g., reconstruction loss). The paper's claims are not well-supported without these experiments.

**Summary Of The Paper:**

This paper proposes a transformer architecture that can deal with multiple environments, tasks, and state-action spaces for a locomotion environment. Specifically, the transformer takes a description of morphology as a prompt and draws multi-head attention over a sequence of state-action pairs. For training, the paper proposes to imitate experts from simple tasks to complex tasks through a hand-designed curriculum and fine-tune via on-policy RL (PPO). The result shows that pre-training helps, and the proposed architecture can generalize to unseen tasks in few-shot and zero-shot settings.

**Summary Of The Review:**

While this paper makes an attempt to solve an important problem, the presentation of the paper needs to be improved, and the results are not convincing enough to be presented at ICLR.

---

> ### Author Response · Authors · 2022-11-18
> **Author Response to Reviewer Vvnb**
>
> We appreciate your detailed comments and valuable questions.  Below are our replies which might answer your considerations:
>
> > **‘The improvement over the baselines (especially MetaMorph) is not statistically significant (Error bars highly overlap).'**
>
> Our primary contribution is to provide a unified architecture compatible for both offline and online learning, and can be generalized for different body shapes and task conditions. From this aspect, we believe our motivation has been well verified by experiments, in which our ODM can have much better zero-shot performance (Table 4) while have similar or slightly better performance on online/few-shot experiments (Table 2 and 3). Besides numerical evaluations, one needs to re-develop MetaMorph for different body shapes and unseen task while ODM can directly adapt. We have rephrased our contributions and experimental comments to make this argument clearer.
>
> Furthermore, there are also statistical reasons for such phenomena in walker and unimal results in Table 2 and 3. Comparing with relative simpler tasks (hopper, halfcheetah, ant, etc), walker has stochastic process noise, while unimal has dynamically stochastic sampling mechanism among 100 robots at each timestep (more details in Gupta, 2021).
>
> On the other hand, on-policy (online) RLRL (especially when with importance sampling) generally suffers from large variance issue but have the highest performance expectation, which is our main driving algorithm during finetuning (in contrast, offline RL has smaller variance but is more prone to be biased.) [1] Therefore, it is not surprisingly for us to see such results and we believe the return expectation can be used as the main evaluation factor.
>
> Last, our means and variances are calculated from 100 samples in contrast to only 3 samples in the original MetaMorph paper. This further enhances the statistic reliability.
>
> [1] Offline Reinforcement Learning: Tutorial, Review, and Perspectives on Open Problems, Levine, 2020
>
> > **‘There is no ablation for the use of curriculum in the pre-training phase, the use of morphology prompt, and the use of each loss term'**
>
> Thanks for the valuable suggestions. Ablation of pretraining phase (as well as finetuning) was in the original version of supplemental materials. To make this analysis more evident, we have moved it to the formal paper (Table 5). The DT and PPO baselines are equivalent or close to ablations of PPO/Imitation loss terms, and corresponding explanation is added in revised version of paper. We have conducted ablation experiments on prompt and curriculum and also add these results in the revised version. We believe these results and analysis could verify effectiveness of model components.
>
> > **‘Why is it called Morphology-aware "Auto-Encoder" when there is no reconstruction loss?'**
>
> We are sorry for the misleading name since there is no reconstruction loss in our approach. We have rephrased the ‘morphology-aware auto-encoder’ to ‘morphology-aware encoder’ to avoid the potential misunderstanding.
>
> > **‘Lprediction seems like a reconstruction loss. Why would it encourage the agent to learn a "world model"'**
>
> There might be some clarification needed. In our work, we want to mimic the determination status of natural agents, with ground truth of future observations and actions are not available when making the decision. Therefore, the backbone of our model is a casual transformer with **time casual mask** (Fig 2). On the contrary, reconstruction is usually employed in sequential modeling when there is **no** casual time mask. One example of reconstruction might be the case that given current-timestep observation (maybe also current action and previous info, etc) and reconstruct the current-timestep observation through the decoder, i.e $f(s_t, a_t) \rightarrow s_t$.
>
> However, our purpose is to make the model know what the **next-timestep** observation is (see the alignment between bottom tokens and top heads in Fig 2) at current timestep of $t$, but the time casual mask inhibits the model knowing the ground truth of $s_{t+1}$. Therefore, $f(s_t, a_t) \rightarrow s_{t+1}$ is more close to prediction than reconstruction. This matches the idea of ‘world model’, which is, agent knows how the world proceeds and what would be resulted from current action. This is a well-accepted notation in prior RL works [1,2]
>
>  [1] Recurrent World Models Facilitate Policy Evolution, NIPS 2018
>
> [2] DayDreamer: World Models for Physical Robot Learning, CoRL 2022

---

> ### Author Response · Authors · 2022-11-18
> **Author Response 2 to Reviewer Vvnb**
>
>
> > **‘Although the particular architecture is new, the idea of learning a universal architecture that can handle multiple morphologies and tasks through supervised learning (e.g., GATO) and reinforcement learning (e.g., MetaMorph) has been explored’**
>
> We are trying to propose a novel approach to learn universal policy on different agent body morphologies by designing a unified model architecture. GATO (and DT, TT, etc) is a great work on general purpose intelligence instead of the embodied intelligence, since it does not have specific configurations on body morphology. Furthermore, they belong to offline learning methods with performances suffering from distribution shift between offline dataset and online environment. Our results show that DT fails in our few-shot and zero-shot experiments.
>
> MetaMorph, on the other hand, considers the agent morphology but only designed for the unimal environment and is not generalized enough for unseen body shapes and tasks. Also, unlike GATO and DT, MetaMorph does not study the time-dependency which could be a problem for more complicated sequential problems. We believe we make significant progress from MetaMorph by introducing the time sequence modeling and explicit multi-task-specific morphology-aware encoders.
>
> In short, our ODM studies the agent embodied behavior, have much better online performance than GATO/DT, while is more generalized than MetaMorph for new type of body shapes and unseen tasks. We have added a Related Works Section to make these statements clearer.
>
>
> - To summarize,
>
> Upon these replies we believe the main concern of Reviewer has been answered or could be further clarified. Reviewer’s reconsideration on the score would be highly appreciated.

---

### Official Review · Reviewer_o4Ch · 2022-10-24

**Confidence:** 4
**Correctness:** 3
**Technical Novelty And Significance:** 3
**Empirical Novelty And Significance:** 3
**Recommendation:** 6

**Clarity, Quality, Novelty And Reproducibility:**


Objectives of the paper is well described with clarity. Formulas and architectures are helpful to understand the proposed architecture. This paper proposes new learning architecture and has enough experiments and comparisons with baselines.
This paper has no codes available.

**Strength And Weaknesses:**

The strength of this paper is as follows.
1 new idea of a time-morphology transformer, ODM,  that takes advantage of both time and morphology dependency for a general purpose policy.
2 new idea of learning from others and on-policy RL using curriculum learning and PPO
3 generalization performance for different environments
Also, this paper shows improved learned motions rationality, smoothness, and agility to support claims.


**Summary Of The Paper:**

This paper introduces Online Decision MetaMorphFormer (ODM) to learn a universal body control policy in arbitrary body shapes, environments and tasks.
The paper proposes a time-morphology transformer architecture which take advantage of both time and morphology dependency for a general purpose policy. Also, motivated by cognitive and behavioral psychology, this paper combines learning from others based on curriculum learning with on-policy RL. The experiments show that ODM is adaptive in the real world and also learns generalized knowledge, such as motions with rationality, smoothness, and agility.

**Summary Of The Review:**

This paper introduces new learning architecture called Online Decision MetaMorphFormer (ODM) to learn a universal body control policy in arbitrary body shapes, environments and tasks.
Objectives of the paper is clear, and experiments show better performance.
With a time-morphology transformer, earned motions have more rationality, smoothness, and agility as well

---

> ### Author Response · Authors · 2022-11-18
> **Author Response to Reviewer o4Ch**
>
> Thanks a lot for your positive assessment of our work and its major contribution.
>
> The code is provided and can be obtained from our website https://baimaxishi.github.io/. We will also include the code in our supplemental materials.

---

### Official Review · Reviewer_UikL · 2022-10-24

**Confidence:** 3
**Correctness:** 3
**Technical Novelty And Significance:** 2
**Empirical Novelty And Significance:** 3
**Recommendation:** 6

**Clarity, Quality, Novelty And Reproducibility:**

The paper is quite well structured but needs refinement. The code, hyperparameter, and environment details for reproducibility are provided in a hyperlink https://baimaxishi.github.io/ and the appendix.


### Clarification questions:
- Writing in section 3.1 is a little unclear, in terms of what aspects are differently modeled than Gupta et al 2022.
- Line 4 in Algorithm 1: switch between 6 ~~environments~~ body shapes?
- Fig 2 is unclear and has typos. How can the same encoder/decoder (denoted by the same color) be used for state and action representation, when they have different dimensional inputs? Eq 9 and 10 are not evident in the figure.
 ~~pertaining~~ pretraining, ~~atenion~~ attention.
- Fig 3, what is the x-axis?

#### References
----
Agrim Gupta, Silvio Savarese, Surya Ganguli, and Li Fei-Fei. Embodied intelligence via learning and evolution. Nature Communications, 12, 2021. URL https://doi.org/10.1038/ s41467-021-25874-z.

**Strength And Weaknesses:**

## Strengths
The paper presents an integrated pipeline of pre-training from offline trajectory dataset & finetuning with RL for different body shapes. The state-action representation based on Gupta et al 2022, handles proprioceptive and exteroceptive observations and continuous-valued torques for each DoF in every joint of the agent. While existing papers have used such a general framework before, this work demonstrates how it can pre-train with supervised learning and fine-tune with RL.
The evaluation highlights how a (1) pure online method like MetaMorph could take some time to improve the returns as compared to pretrained ODM in unimal and walker, (2) a pure offline method like DT can not adapt to changing body shape to unimal.


## Weakness
The pretraining with supervision and finetuning with RL is quite common and may work in certain environments, and the work showcases this sensitivity in Fig 4. However, the work does not introduce any specific inductive bias in modeling or loss to handle the potentially conflicting learning signal in offline pretraining and online RL. The concern is that the shown results for combined pretraining + finetuning are currently highly sensitive to training tricks and schedules used and might not generalize.

How does the two-phase training for ODM compare to MetaMorph in terms of the wall clock time? The claim that ODM starts better in Fig. 4 is unclear as the pretraining time is unknown.

The section on limitations or concrete future directions is missing.

**Summary Of The Paper:**

The paper proposes Online Decision MetaMorphformer (ODM), a pipeline based on transformer structure to encode the morphology and temporal proprioceptive & exteroceptive information to learn from offline trajectories and finetune from environment interaction. This policy shows generalization in unseen settings with uneven terrain and obstacles, as well as few-shot finetuning. The state is modeled in terms of proprioceptive and exteroceptive observations and the actions are torques for each DOF of the joint. The trajectory is passed through a casual transformer to predict the subsequent next state and action per timestep. This is pre-trained to minimize MSE between the states and action predictions with respect to a dataset, sorted from the easiest to the hardest task for curriculum learning.8 different body shapes, interactive environments, and few-shot & zero-shot tasks are considered to learn and evaluate the policy. The proposed approach (ODM) is compared to MetaMorph, DT, PPO, and random baseline.



**Summary Of The Review:**

Overall, the paper presents an interesting integration of supervised pre-training and online RL fine-tuning over morphologically different agent trajectories and demonstrates few-shot and zero-shot transfer to unseen shapes and environments. While the ideas for state-action representation, transformer-based modeling, and two-phase training paradigm existed in common knowledge, the paper presents an interesting evaluation with pure offline (DT) and online (MetaMorph) variants to address the same problem. It opens a broader research question to the community on how to reduce the conflict between the learning signal between two training phases across differing morphological shapes and environments.

---

> ### Author Response · Authors · 2022-11-18
> **Author Response to Reviewer UikL**
>
> Thanks a lot for your positive assessment of our work and its potential contribution.  According to your remaining considerations, here are our detailed replies:
>
> - **'the work does not introduce any specific inductive bias in modeling or loss to handle the potentially conflicting learning signal in offline pretraining and online RL. The concern is that the shown results for combined pretraining + finetuning are currently highly sensitive to training tricks and schedules used and might not generalize.'**
>
> This is a very good question which points out the potential conflict between offline and online learning signals. We do find there are some performance conflicts when training switches from offline to online phase, as exhibited in Fig 4. One possibility is this issue could be improved by further hyperparameter tuning (e.g. better warmup of online learning rate). Our argument is that this work is focus on introducing a unified transformer architecture which is both compatible with auto-regressive offline learning, as well as the actor-critic-based online learning (the first time), while further hyperparameter tuning is unfortunately somewhat beyond our current scope, due to effort and page limits. Although this limitation, our multi-task setting and few/zero-shot tests can in somewhat verify the generalization ability of our two-phase learning mechanism.
>
> On the other hand, we cannot rule out the possibility that there are conflicts between offline and online learning knowledge based upon current model architecture. We have added a new Discussion Section with this consideration included. We believe it is valuable to broadcast this issue to the community and calls for future interests and solutions.
>
> - **'How does the two-phase training for ODM compare to MetaMorph in terms of the wall clock time? The claim that ODM starts better in Fig. 4 is unclear as the pretraining time is unknown'**
>
> Comparing our total learning time (pretraining + finetuning) with a pure online method like MetaMorph is trivial since the pretraining phase mainly aims to take advantage of more data and accumulate more task/domain knowledge.  Once pretraining is completed, a stable checkpoint can be repeatedly loaded and reused for different downstream tasks while MetaMorph needs to re-develop for a new task. Starting from pretrained result, finetuning becomes more efficient on learning speed and sampling efficiency, as indicated by Fig 4.
>
> Nevertheless, we agree that the time comparison is not clear enough in the original version of paper. We have added the x-axis in Figure 4 and 5 and make the clear statement that our total learning time is longer than MetaMorph but finetuning is shorter.
>
> - **For clarification questions:**
>
> We have re-organized preliminary and method sections to make notations and definitions clearer. Fix typos and add x-axis in Fig 3 and 4.
>
> Colors of model components in Fig 2 are used to label their functionality but not dimensionality. Morphology-aware Encoders are part of backbone (and task-agnostic) since we already map original observations and actions into the same dimensional (dim=e) latent space using task-specific tokenizers. We have moved Projectors (which outputs observations and actions) outside the backbone since they do have different dimensionality (task-specific). Thanks for reviewer’s kindly reminder.

---

### Official Review · Reviewer_t6eV · 2022-10-25

**Confidence:** 3
**Correctness:** 3
**Technical Novelty And Significance:** 2
**Empirical Novelty And Significance:** 2
**Recommendation:** 3

**Clarity, Quality, Novelty And Reproducibility:**

Clarity:
- 2/5

Quality
3/5

Novelty
1/5

Reproducibility
3/5

**Strength And Weaknesses:**

Strengths:
- The proposal is complete and can be trained on various sim environments
- Apparently able to learn from multi robot data

Weaknesses:
- Tested on sim only. Online learning is too slow to be practical in real.
- The paper skips over a lot of prior work in online learning, relevant work section is pretty basic and boiler plate
- Visual examination of frames is not exactly relevant. The dynamics of a human and humanoid may be different in sim to make difference joint configurations optimal

**Summary Of The Paper:**

This paper proposes an architecture and method for online training for robotic datasets

**Summary Of The Review:**

The paper proposes a mechanism to learn from multirobot learning and tests on sim environment in an online learning fashion.

---

> ### Author Response · Authors · 2022-11-18
> **Author Response to Reviewer t6eV**
>
> We appreciate the reviewer’s detailed comment.  The feedback is very helpful but the following questions can still be clarified. Here are our detailed replies:
>
> - **‘Tested on sim only. Online learning is too slow to be practical in real.’**
>
> Behavior learning on sim is a quite generally setup in the RL field, since one tries to solve a black box problem, while simulator provides well-defined state, action and reward spaces, and facilitates learning process. Therefore, almost all the classical RL (A3C, PPO, TRPO), as well as recent advances of offline RL (CQL, BCQ, IQL) and transformer-based RL (DT, TT, GATO) study on sim only. Well-studied RL methodologies on sim can then be successfully transferred to the real world application with the same problem setup (sim2real). Therefore, we believe sim is also a well-defined platform for current state of this work.
>
> Within the RL scope, online learning generally has better performance, although also suffers slower learning speed Current advances on transformer-based RL uses offline learning which is faster, but has apparently worse performance than online RL (For example, Table 2 and 3 in the DT paper (Nips Chen 2021) has metrics small than 100, which are normalized by 100, the 'online expert'). Although such works are intuitive and inspiring (when working on large dataset and large model; multi-task / multi-modal), in this work we determine to endorse online learning paradigm with the transformer-based RL, to keep or even outperform the sota performance.
>
> - **‘The paper skips over a lot of prior work in online learning, relevant work section is pretty basic and boiler plate.’**
>
> Thanks for kindly reminding. We have re-organize the paper and add the section of Prior Works, to better illustrate our novelty.
>
> - **‘Visual examination of frames is not exactly relevant. The dynamics of a human and humanoid may be different in sim to make difference joint configurations optimal.’**
>
> We do believe visualization of humanoid is meaningful because what we are studying is to learn a smart humanoid in a simulated environment (as explained in the first question).  Such frames also provide intuitive proofs with knowledge transferred from the real world. (For example, it should be reasonable to expect a humanoid can walk better with its two legs interleaved move, instead of keep moving only one leg). From this aspect, we believe our visual results provide another manner to our sota performance of humanoid.

---

### Decision · Program_Chairs · 2023-01-20

**Decision:**

Reject

**Justification For Why Not Higher Score:**

Two reviewers indicate reject and two indicate a weak accept.  Reviewer VVnb is senior and provided an insightful review.  After examination of the paper, the area chair believes that the paper in its current form does not present its ideas with adequate clarity for publication.

**Justification For Why Not Lower Score:**

N/A

**Metareview: Summary, Strengths And Weaknesses:**

Summary:

This paper proposes a transformer architecture (ODM) that can deal with multiple environments, tasks, and state-action spaces for a locomotion environment. Specifically, the transformer takes a description of morphology as a prompt and draws multi-head attention over a sequence of state-action pairs. For training, the paper proposes to imitate experts from simple tasks to complex tasks through a hand-designed curriculum and fine-tune via on-policy RL (PPO). The result shows that pre-training helps, and the proposed architecture can generalize to unseen tasks in few-shot and zero-shot settings.

Strengths:

The reviewers appreciated the novelty of the algorithm for handling multiple morphologies.  The author response addressed some misunderstandings on the application of past works to multiple morphologies (connections to MetaMorph and Gato).

Weaknesses:

The paper retains a poor attention to presentation detail, and contains both simple typos (casual/causal) and loose technical writing (reviewers Vvnb, UikL).  These errors impede communication with the reader, and significantly reduce the potential contribution of the work.
Reviewer Vvnb commented "The improvement over the baselines (especially MetaMorph) is not statistically significant (Error bars highly overlap)."  This a big issue, and speaks to a lack of clarity in communication, despite the author's response to this comment.

The experimental results are also troublesome in several ways.  The tables bold differences that do not appear to be statistically significant.  The use of PPO seems to be a weak baseline in comparison to SAC.  Even more troubling is the fact that random seeds are only mentioned in testing and not in training (presumably only one seed).  There are deep flaws either in the descriptions of the experiments in the paper, or in the experiments themselves.